



# Investigating Lab-scaled Offshore Wind Aerodynamic Testing Failure and Developing Solutions for Early Anomaly Detections

Yuksel R. Alkarem[1], Ian Ammerman[2], Kimberly Huguenard[1], Richard W. Kimball[2], Babak Hejrati[2], Amrit Verma[2], Amir R. Nejad[3], Reza Hashemi[4], and Stephan Grilli[4]

[1]Civil and Environmental Engineering Department, University of Maine, 35 Flagstaff Road, Orono, Maine 04469, USA
[2]Mechanical Engineering Department, University of Maine, 35 Flagstaff Road, Orono, Maine 04469, USA
[3]Department of Marine Technology, Norwegian University of Science and Technology (NTNU), Jonsvannsveien 82, 7050 Trondheim, Norway
[4]Ocean Engineering, University of Rhode Island, Sheets Building, 15 Receiving Road, Narragansett, Rhode Island 02882, USA

**Correspondence:** Yuksel R. Alkarem (yuksel.alkarem@maine.edu)

**Abstract.** Experimental demonstration of offshore wind components and systems plays an increasingly critical role in advancing the industry. As these systems increase in complexity, the likelihood of human error or software malfunction increases, leading to costly equipment damage. This study examines a laboratory incident which occurred during aerodynamic characterization of a 1:50 scale 5 MW reference wind turbine and presents an efficient early anomaly detection method. During

the experiment, the model generator disengaged causing a catastrophic rotor overspeed and subsequent blade-tower strike. Applying data-driven approach to predict system's dynamics from measurements, this study investigates the potential to enhance reaction time and prediction quality using single- and multi-principal component models. Early system malfunctions are detected with the single-principal component model showing better performance. Sensitivity analyses show gains in reaction time with increasing sample frequency, lending this work in particular to lab-scale systems that operate at high sample rates to

reduce future incidents.

## 1    Introduction

Model scale laboratory testing is a necessity for early development of grid scale on- and offshore wind energy technologies, and recent industry trends have driven increased demand for such testing (Mehlan and Nejad, 2024; Soares-Ramos et al., 2020). In the case of offshore wind energy projects, operation and maintenance costs can amount to a third of the projects life-cycle

cost, often quantified as the levelized cost of energy (LCOE), meaning that the maturity of new technologies for deployment must be improved through small-scale validation and testing (Mehlan and Nejad, 2024; Association, 2009; Leahy et al., 2016; Wang et al., 2022).

To meet this demand, lab-scale turbine systems are designed to match the performance of full-scale offshore commercial wind plants to facilitate accurate coupling of wind turbine dynamics with the hydrodynamic forces on the substructure Fowler

et al. (2023); Kim (2014); Cao et al. (2023). As a consequence of the low-Reynolds wind environment at lab-scale, model turbine blades rely on thin airfoil sections, such as the SD7032, to achieve scaled rotor performance, increasing flexibility



and reducing strength of the blades. Furthermore, due to tight mass considerations, particularly for floating models, system redundancy in the case of equipment malfunctions is not generally designed for Parker (2022). Consequently, lab-scale turbines are highly sensitive pieces of equipment requiring acute care by operators to ensure safe and reliable operation throughout a test campaign.

In experimental testing campaigns, and particularly when testing novel control algorithms, the likelihood of fault events is increased and the consequences of a fault or erroneous command can be severe. These events could fold as consequences of an operator error, erroneous control commands, or instrumentation malfunctions (Peng et al., 2023). Such errors can lead to costly damage to lab equipment, violations of laboratory safety standards, and cause significant project delays; therefore, efforts to develop efficient methods of detecting operational faults are critical to improving the testing process. (Leahy et al., 2016; Lu et al., 2024).

Currently, studies of conceptual vibration-based condition monitoring techniques, such as velocity and acceleration measurements, were proposed for rapid and early online fault detection in commercial-scale systems (Nejad et al., 2018); however, in complex systems composed of multiple interconnected components, vibration signals may originate from multiple sources and the data captured by individual sensors may offer limited insight. Consequently, incorporating multiple data channels is essential for constructing a comprehensive representation of system behavior, though doing so can come at the expense of greater computational cost and the risk of misinterpreting otherwise irrelevant signal data. Such risks can be mitigated through dimension reduction strategies during system pre-processing to improve algorithm efficiency (Dibaj et al., 2022). Other approaches may utilize adaptive filters, such as linear and non-linear Kalman filters for efficient adaptive fault detection, although these strategies can be challenging to implement for increasingly complex systems (Zhou and Zhu, 2023; Le and Matunaga, 2014; Ammerman et al., 2024). Online data-driven prediction and forecast models have also been used in the past to detect changes in the systems' state or any environmental conditions (Dibaj et al., 2022; Alkarem et al., 2024, 2023).

These and similar methods can also be applied to lab-scale models, with the additional caveat that computational efficiency is even more critical. Due to time scaling and typically higher frequencies of motion at lab-scale, fault detection strategies on models must be able to operate quickly and with minimal overhead. To meet this need, pre-trained data-driven approaches offer significant performance benefits over non-linear physics-based models.

The case study in this work comes from a fault incident which occurred during a standard scale model characterization test, wherein the turbine generator disengaged during an experiment, causing the turbine to spin out of control and one of the blades to strike the tower. The resulting damages caused significant delays in the campaign. Using this incident as a real example of the need for online fault detection and mitigation strategies, a data-driven approach was applied to develop an efficient online monitoring system which can detect failures or anomalous behavior before significant system effects are realized, increasing reaction time for operators or enabling automated shutdown procedures to take place.



## 2  Methodology

### 2.1  Experimental setup

The experimental data for this study comes from a wind turbine characterization test performed on a scale model, at the Harold Alfond Ocean Engineering Lab at the University of Maine's Advanced structures and Composites Center. The layout of the experiment is shown in Figure 1a, illustrating the arrangement of the wind machine and turbine model. During the experiments, the turbine was controlled and monitored by 1 or 2 test operators stationed to the side of the basin, via a data acquisition system (DAQ) based on the National Instruments cRIO platform. Figure 1b shows the installed experimental turbine before testing

began. To properly characterize the turbine's aerodynamic performance, it was installed in a fixed configuration within the wind field. Cross-bracing was installed to keep the turbine tower and mounting surface rigid during the test to target rotor performance only.

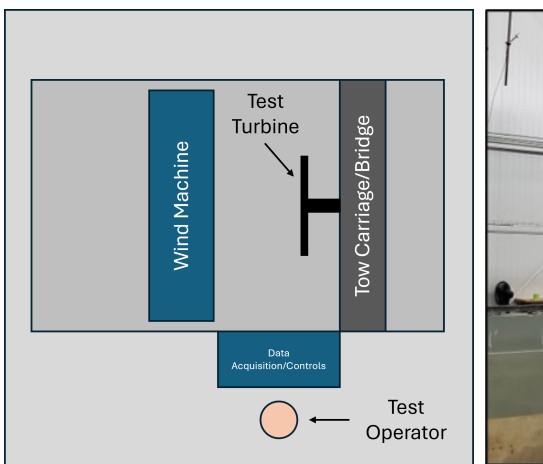 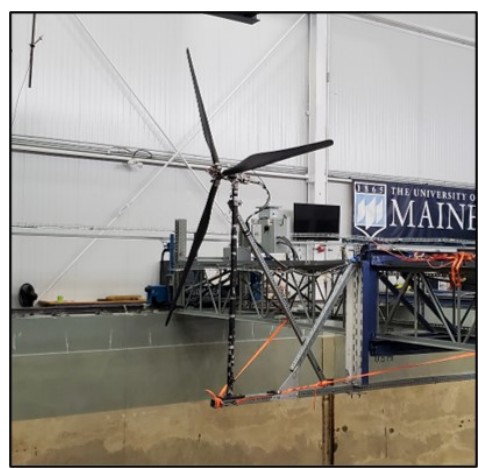

(a) Basin layout for scaled turbine characteriza-   (b) Photograph of the experimental test turbine
tion experiment.                                      installed in the basin.

**Figure 1.** Experimental test setup overview (a) and image of installed turbine (b).

To fully characterize the rotor, experiments were performed at various wind speed/RPM pairs. Each experiment used a previously generated setpoint file to cycle through blade pitch setpoints. Figure 2 shows blade pitch (2a) and rotor thrust (2b)

from one of the experiments. Results from these tests were then used to form rotor performance surfaces for future experiment design.

### 2.2  Failure incident

During one of the characterization runs, an operator mistakenly triggered an emergency stop on the turbine generator. As a result, the rotor began accelerating unrestricted until a blade strike occurred with the tower and the wind generator could be



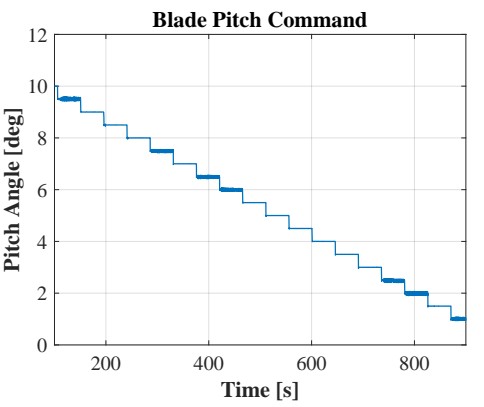
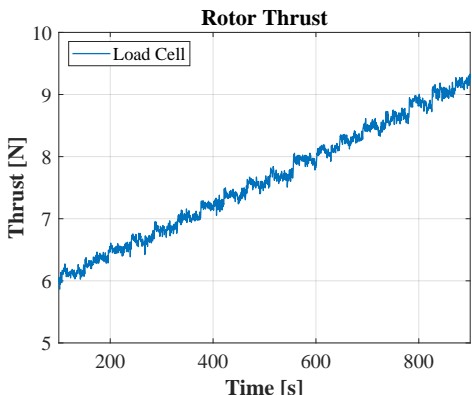

(a) Blade pitch schedule for sample characterization experiment.

(b) Rotor thrust measured for sample characterization experiment.

**Figure 2.** Scheduled blade pitch (a) and measured rotor thrust (b) for sample characterization experiment run.

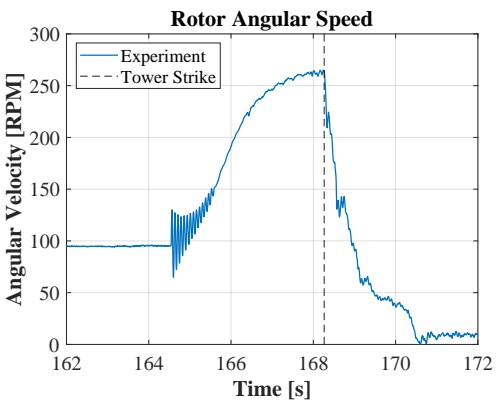
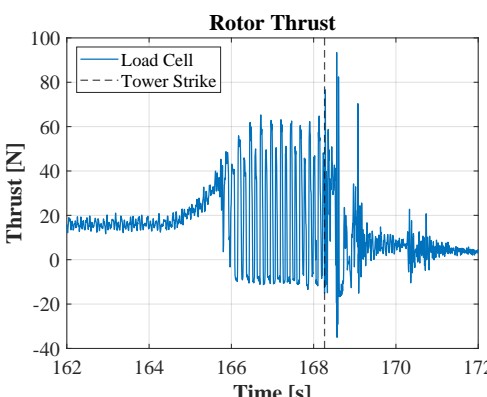

(a) Basin layout for scaled turbine characterization experiment.

(b) Photograph of the experimental test turbine installed in the basin.

**Figure 3.** Measured rotor RPM (a) and thrust (b) during blade strike incident.

shut down. Plots of rotor speed and thrust load during the incident are shown in Figure 3, with a vertical line indicating when the blade struck the tower.

## 2.3 Predictive model description

Detecting early signs of anomalies in testing campaigns can be beneficial. It can either provide data where operators can act upon with informative decisions and/or it can be automated to abort the test in case certain thresholds are exceeded. However,
signaling a possibility of an anomaly requires real-time processes of incoming measurements data, which can be best done





using deep machine learning algorithms. Such algorithms indeed make it possible for predictive models to be trained on certain healthy data then, when processing upcoming measurements in the lab during similar testing, to provide predictions or forecast of a certain state of the system.

Accidents with lab equipment can be costly and labor intensive and can cause delays. to mitigate such incidents, we propose
an early anomaly detection model to improve response times and reduce human error. To this effect, a multi-step, multivariate Long Short-Term Memory (MLSTM) model — a type of recurrent neural network (RNN) designated to address the vanishing gradient issue that traditionally prevents models to capture long-term dependencies — was developed and trained on data from a healthy aerodynamic characterization tests with similar wind speeds. When an anomaly occurs, the error between the predicted signal and the measured signal increase which can be used to inform the operator of such incident. The model parameters were
initially estimated intuitively, but these could be further refined for enhanced predictive accuracy.

## 2.4   Anomaly detection over the span of multiple channels

In complex systems such as offshore wind testing, there are numerous measurements and data channels, which can be used to understand the overall behavior of the system. However, for anomaly detection purposes, it can be overwhelming and computationally expensive to manually and in real-time search the data space for deviation in measured data. The operator
might not have sufficient time to abort the test before the anomaly becomes too consequential. Additionally, an anomaly might not be detectable based on any single data channel to comprehend the full state of the system. Therefore, the predictive model must be based on multiple data channels related to the test being conducted while providing the operator with a single, concise, anomaly detection capability based on the most relevant information. To accomplish that, principal component analysis (PCA) was carried out. A PCA creates combinations of variables that explain the largest amount of variance in the data.
Prior to performing this analysis, earlier data reported by the data acquisition system is cleaned to remove idle measurements or non-numeric values. Then, the data are prepared for the PCA, by standardizing them, using their mean and standard deviation, which ensures that all channels (features) are on the same scale to prevent features with larger ranges dominate. The covariance matrix is then computed for the standarized variables, which is then rotated to become a diagonal matrix, with transformed variable, a.k.a. principal components (PCs), ranked from those describing the largest to the lowest fraction of the
total variance. One can finally decide which components to retain for the purpose of reducing the problem dimensionality, which still explaining the largest possible amount of variance.

The PCs, which are expressed as weighted sums of the earlier variables, were then used to train the RNN model(s) that will later be used for prediction. As new data is acquired, it is transofmed/projected onto the same PCs that were used in training the models. For the purpose of anomaly detection, the mean absolute error (MAE) is computed between measurements and
predictions from the RNN models, and the error derivative is calculated, to estimate rapid fluctuations in the quality of the predictions. An anomaly alert is reported to the operator when certain anomalous conditions are met. In this research, we investigate conditions when both the error and its derivative were crossing certain thresholds. This procedure is illustrated in Figure 4 and is explained in section 2.6.





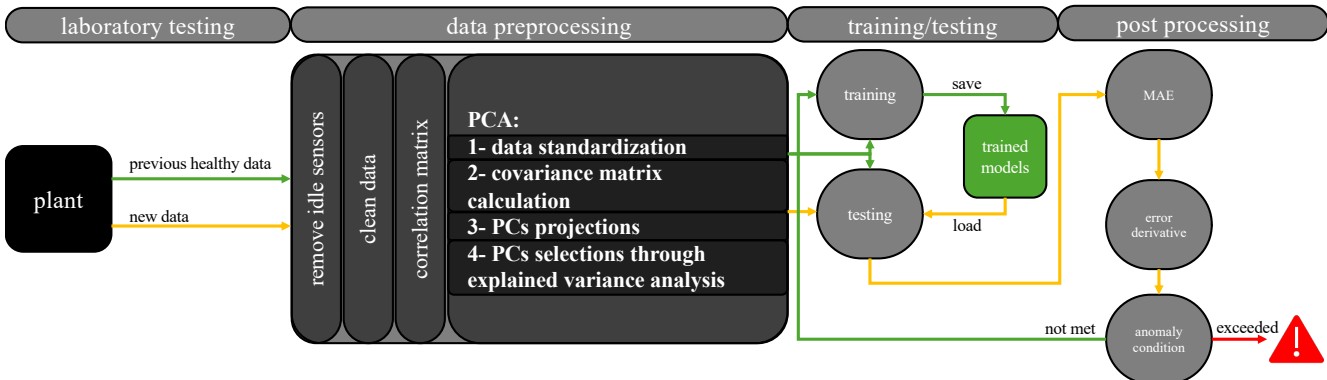

**Figure 4.** Flowchart describing data stream, data preprocessing, training the MLSTM model and using it for anomaly detection.

## 2.5  Principal component selection

Two models developed vary in their projected principal component selection. The projected PC results from training data are presented in Figure 5. The first model compresses the data by retaining only the first PC; it is therefore named '1PC'. The second model selects the group of (M) PCs that cumulatively explain 90% of the total variance, thus only neglecting the remaining 10%; this model is hence called 'MPC'.

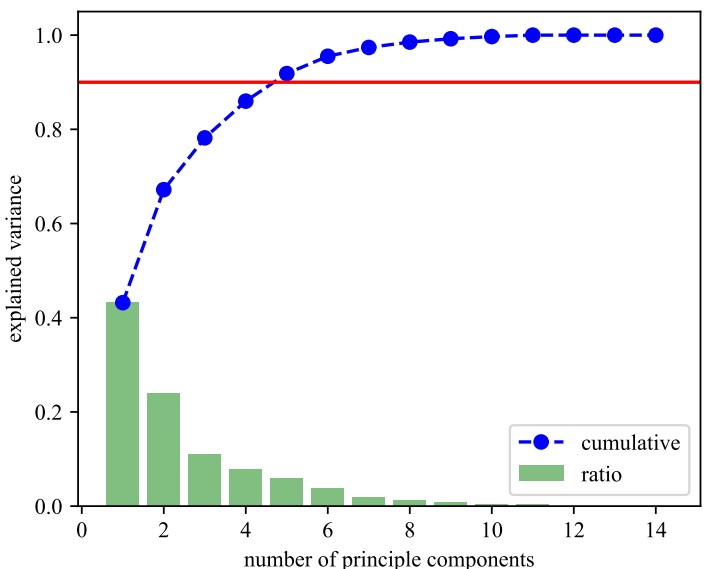

**Figure 5.** Explained variance ratio and cumulative of all principal components.



## 2.6 Error and error derivative thresholds selection for single/multiple PCs

The histogram of the derivative of the error between the trained model and the training data for the 1PC and MPC models are shown in Figure 6. Inspired by the work of Dibaj et al. (Dibaj et al., 2024), the thresholds were selected to be the highest values in the histogram for the training data error. This will later be used to assess whether or not the predictive model is diverging from the measured data during the testing/anomaly detection stage.

## 2.7 Performance metrics

The overall accuracy of the model(s) measured by a single score that combines precision and recall in its calculation (Miele et al., 2022; Wang et al., 2019). Precision, $P$, illustrates the proportion of anomalies detected that are true, while recall, $R$, indicates which proportion of true anomalies are detected. They can be computed as:

$$P = \frac{T^+}{T^+ + F^+}, \; R = \frac{T^+}{T^+ + F^-} \tag{1}$$

where $T^+$ represents the count of true positives (the identified anomalies are true), $F^+$ are the count of false positives (i.e.,
for which the identified anomalies are not true), and $F^-$ are the false negatives (i.e., the unidentified true anomalies). These contribute to an overall $FI$ score that ranges between 0 and 1 with 1 being a perfectly precise model and is expressed as:

$$FI = 2 \times \frac{P \times R}{P + R}. \tag{2}$$

## 3 Results

### 3.1 Pre-strike anomaly detection

During one of the high wind speed tests for the aerodynamic characterization of a 1:50 scaled wind turbine, an erroneous activation of the emergency stop led to the shutdown of the generator. This caused the rotor to spin rapidly increasing thrust forces on the blades, causing them to bend more. Within less than four seconds, one of the blades struck the tower, resulting in severe damage as shown in figure 7.

The model used to detect an anomaly is based on a double condition criteria; both the error and its derivative must exceed
the training error thresholds. This was shown to provide the most accurate model (from a sensitivity study carried out and presented below). The trained model tested against the run where the anomaly took place. The evolution of the error derivative of the first component for 1PC model is shown in Figure 8a and the average error derivative of the first five components for MPC model in Figure 8b. The vertical black dashed line is the inception of the anomaly and the vertical red dashed line is the earliest detection by the models; note, as it is impossible to detect an actual anomaly before its inception, the red line can only
exist on the right side of the black line. Before the anomaly, both models have error derivatives below the threshold, indicating an accurate representation of a healthy system befxore the anomaly. The red dots are when anomalous conditions were met. In the present case, we see that the 1PC model able to detect the anomaly earlier than the MPC model as the error derivative





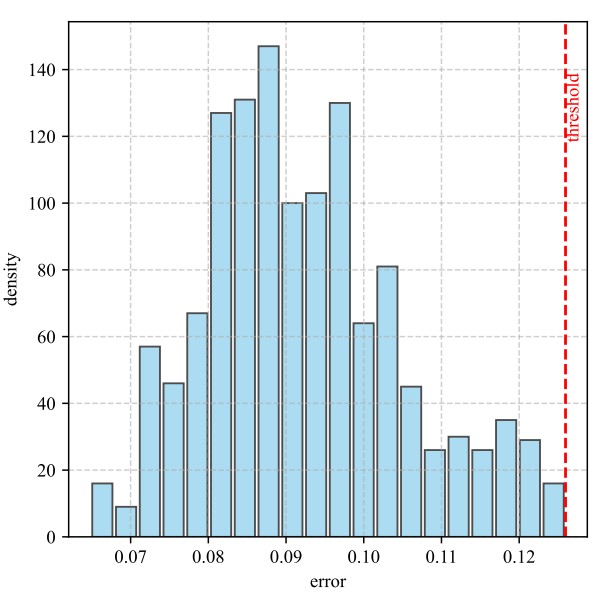

(a) Error histogram in 1PC model.

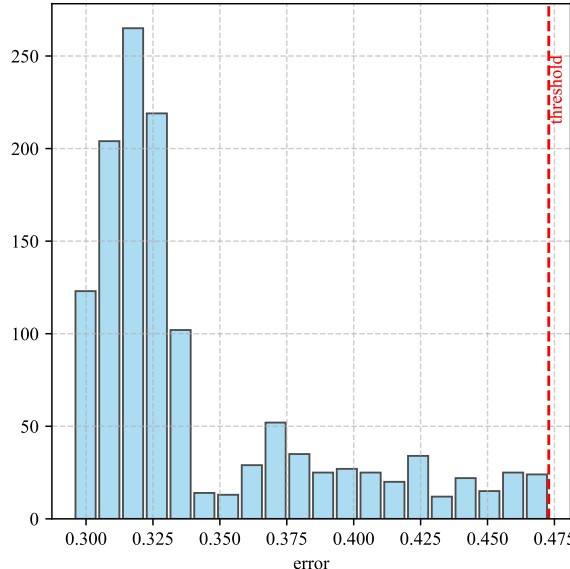

(b) Error histogram in MPC model.

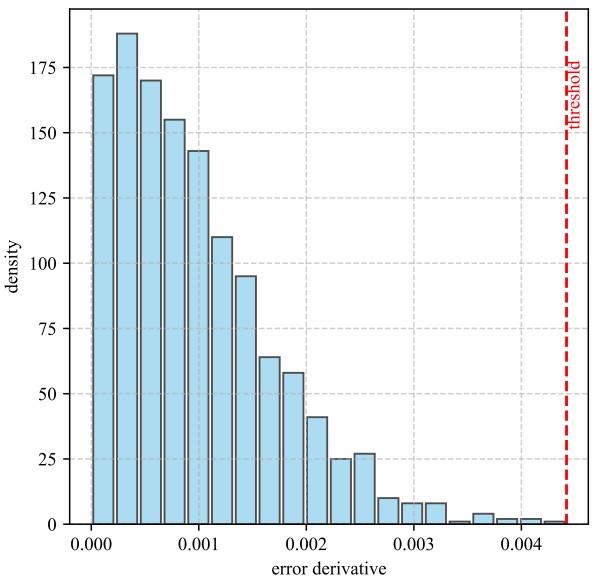

(c) Error derivative histogram in 1PC model.

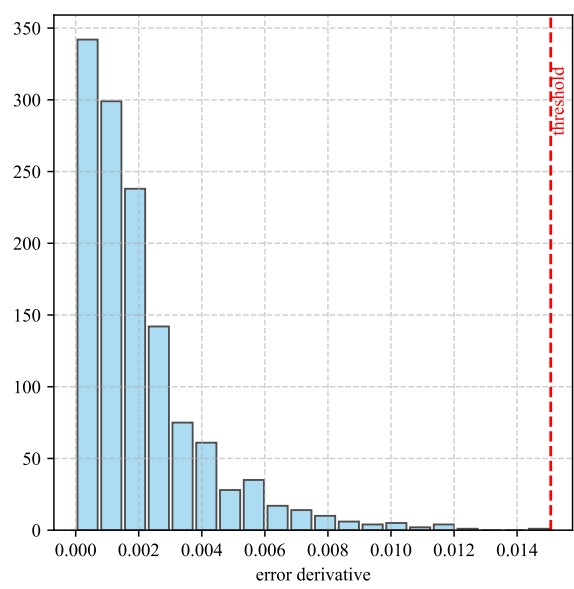

(d) Error derivative histogram in MPC model.

**Figure 6.** Histograms of the (a) error for 1PC, (b) error derivative for 1PC, (c) error for MPC, and (d) error derivative for MPC throughout the training data and the maximum error/error derivative as the selected threshold.



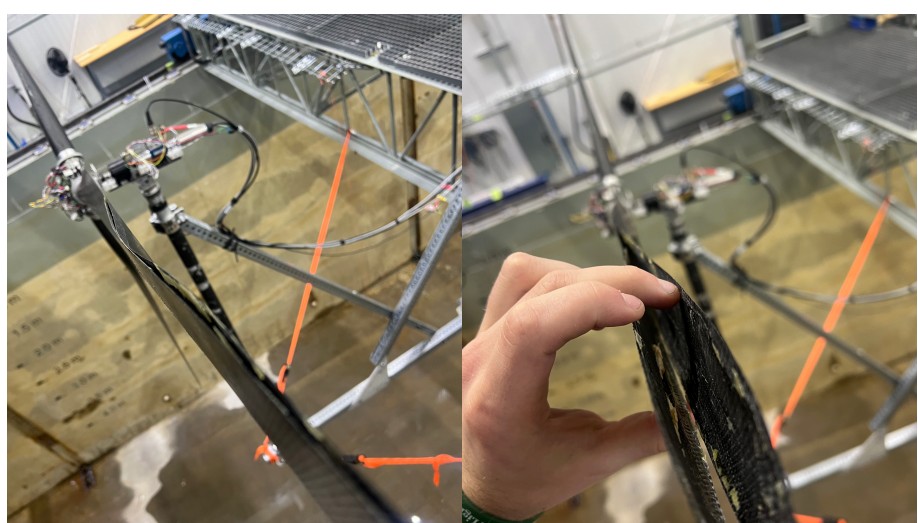

**Figure 7.** Blade damage after blade-tower collision due to high thrust forces.

breaches the threshold earlier. The reason of having a drop in the error derivative after an anomaly has started is because the error at that time reaches its peak and reverses down as the prediction model works to acquire to the available historical data.

**3.2 Variation of anomaly detection criteria combination to the accuracy of the models**

Three combination of anomaly detection criteria and their effects on $FI$ score were investigated. Table 1 and Table 2 provide a description of these variations for the 1PC and MPC models, respectively. The $E$ and $\Delta E$ symbols refer to the error and the error derivative threshold exceeding criteria, respectively tested under normal healthy run operations (HR) or anomaly runs (AR). The combinations studied were 1) sole derivative, 2) a combination of absolute error value and its derivative must be met, and 3) either the error or its derivative must be exceeded for the anomaly to be triggered. The precision and recall as well as the $FI$ score are presented. As the models were trained on healthy data, when being tested under other healthy data (with slight variations in blade pitch angle being tested), the HR, since it does not contain any anomalies, reports zero $T^+$ and $F^-$ values, but a non-zero $F^+$ value for $\Delta E$ and $\Delta E|E$ combinations because some normal samples were misclassified as anomalies. The $\Delta E \& E$ condition, however, does not trigger false anomalies and it also shows the highest $FI$ score. The MPC model shows similar patterns but with lower $FI$ scores.



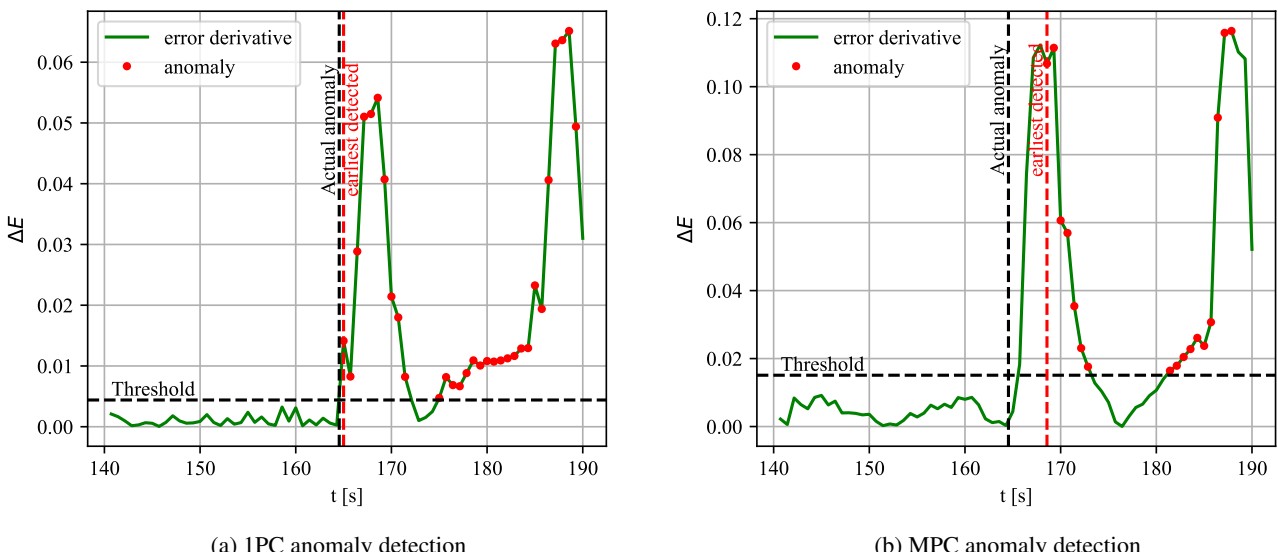

(a) 1PC anomaly detection

(b) MPC anomaly detection

**Figure 8.** Anomaly detection during the run when anomaly occurred and model detection response using 1PC (a) and MPC (b) modes.

**Table 1.** 1PC quality of anomaly prediction quantification.

| 1PC | | $T^+$ | $F^+$ | $F^-$ | Precision | Recall | FI Score |
|---|---|---|---|---|---|---|---|
| **ΔE** | **AR** | 33 | 32 | 4 | 0.452 | 0.892 | 0.600 |
| | **HR** | 0 | 8 | 0 | | | |
| **ΔE & E** | **AR** | 32 | 15 | 5 | 0.681 | 0.865 | 0.762 |
| | **HR** | 0 | 0 | 0 | | | |
| **ΔE | E** | **AR** | 37 | 149 | 0 | 0.188 | 1.000 | 0.316 |
| | **HR** | 0 | 11 | 0 | | | |

## 3.3 Sensitivity study of sampling frequency to anomaly detection delay, simulation speed, and accuracy

The two models were tested FOR the test run that includes the anomaly and the blade-tower strike. The anomaly to strike duration about 3.7 seconds, and is shown as a red dashed line in Figure 9a. Any delay less than that duration is considered a successful anomaly detection and the earlier the better. As is apparent in the results, the 1PC model the superior option as it provides: 1) a shorter delay in anomaly detection, 2) a faster computation due to the fact that only a single PC needs to be predicted, compared to 5 PCs in the counterpart MPC model, as seen in Figure 9b, and 3) a higher $FI$ score across the sampling frequencies tested as illustrated in Figure 9c.


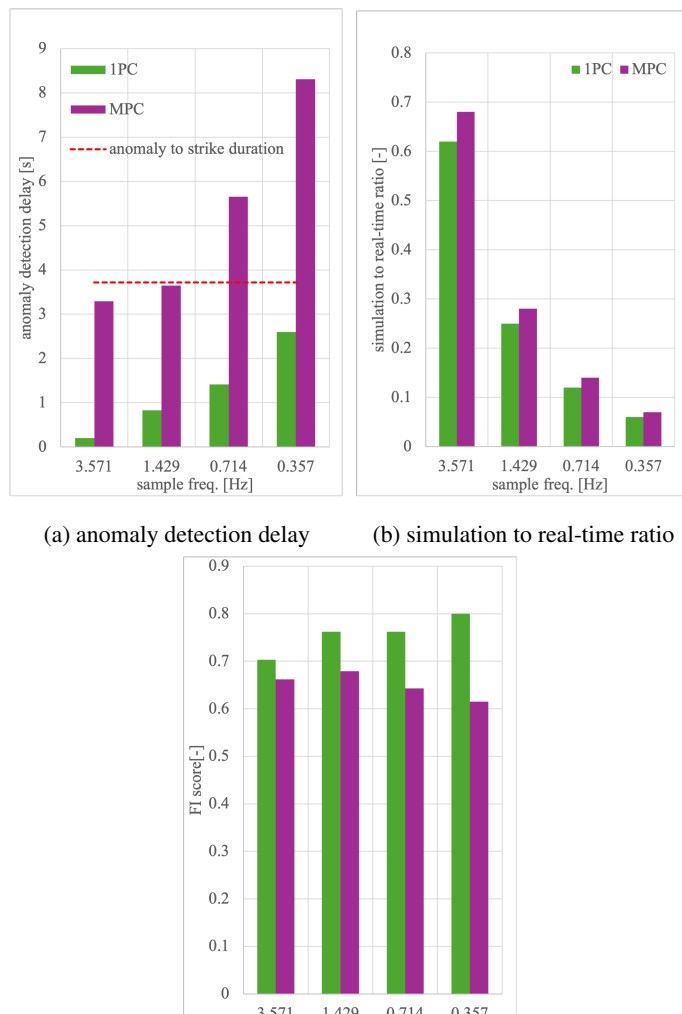

(a) anomaly detection delay     (b) simulation to real-time ratio

(c) FI scores

**Figure 9.** Sensitivity study of sampling frequency on (a) anomaly detection delay, (b) simulation nto real-time ratio, and (3) FI score for both 1PC and MPC models.





**Table 2.** MPC quality of anomaly prediction quantification.

| MPC | | $T^+$ | $F^+$ | $F^-$ | Precision | Recall | FI Score |
|---|---|---|---|---|---|---|---|
| **ΔE** | **AR** | 25 | 10 | 12 | 0.510 | 0.676 | 0.581 |
| | **HR** | 0 | 14 | 0 | | | |
| **ΔE & E** | **AR** | 19 | 0 | 18 | 1.000 | 0.514 | 0.679 |
| | **HR** | 0 | 0 | 0 | | | |
| **ΔE \| E** | **AR** | 35 | 10 | 2 | 0.574 | 0.946 | 0.714 |
| | **HR** | 0 | 16 | 0 | | | |

## 4  Discussion

Results shows that the error and error derivative criteria ($\Delta E \& E$) the better option in terms of anomaly response time and
accuracy of the response. Applying PCA on the data before training the model(s) helps reducing the problem size, which
can be very helpful when there are many data channels that make it extremely laborious to do intensive search to detect
an anomaly. When only considering the first principal component (1PC), the anomaly detection quality surpasses that of a
multiple components model such as with the MPC model by 12% under the $\Delta E \& E$ anomaly criteria. Additionally, the 1PC
model shows a faster response to anomaly and is computationally more efficient, rendering it the superior option between the
two considered here. The 1PC response time can range from a mere 6% to 70% of the duration between the inception of the
anomaly and the blade-tower strike based on the sampling frequency as shown in Figure 9a. The MPC range under the same
sampling frequency variations were between 89% and 225%. Which means that, if the sampling frequency is not small enough,
the anomaly can have detrimental effects before it is even detected if multiple-PC models were being used for that detection.
With the two models tested with healthy data under different operating conditions, the number of anomalies detected was small.
If the $\Delta E \& E$ anomaly conditions were selected, both false positives and negatives counts for healthy data were zero.

As for the incident dataset, since the experiment has also different operating conditions, the error was high. Therefore, using
the error derivative as an additional criterion helps with detecting the true positives. This is an important aspect of a good
anomaly detection model because most of the time, the anomaly will most likely occur under conditions that have not been
seen before.

It is eventually up to the developer/operator to gather certain amount of anomaly points before activating an alerting system
or before acting upon it to limit disturbance to the main testing campaign. It could also be developed such that the model can
trace back to which of the channels contributing to this anomaly based on the correlation matrix calculated during the PC
analysis.





# 5 Conclusions

This research highlights the potential for multi-variate long short term memory (MLSTM)-based model to enhance the safety and reliability of on- and offshore wind experimental campaigns at lab-scale, with the possibility of extension to ocean and commercial scale deployments. After proper data preprocessing, scaling, and their covariance computed to project onto principal components, the healthy data were trained and two criteria for anomaly detection to occur was studied including threshold exceedance of error or error derivative or a combination of both. When combining both criteria, the anomaly detection was
observed to be more accurate and the anomaly response time short. The model with a single principal component showed to be the more superior option in terms of response time, computational time, and the prediction quality. When healthy data are tested, no erroneous anomalies were observed for the model that combined both criteria of error and its derivative. This work lays the foundation for future as a proof of concept that such easy to establish techniques can be used as a safety precaution during future campaign testings to reduce human error and equipment malfunction in the laboratory, in particular when innovative
technologies and control strategies are being tested.

# 6 Competing interests

One co-author (Dr. Amir Nejad) is a member of the editorial board of Wind Energy Sicence Journal.

# 7 Author contributions

KH, RK, and BH were responsible for the acquisition of the financial award for the project alongside SG and RH from
the University of Rhode Island and coordinated this research activity. AV and AN initiated a research exchange between the University of Maine and the Norwegian University of Science and Technology that equipped YA and IA with the methodologies and tools used in this research and ultimately led to this publication. BH and IA designed the experiments and IA and YA carried them out. IA managed data collection, while YA with vital input from AN and SG developed the models and performed the analysis. KH, AV, and RH supervised over the experiment and the analysis carried out. YA prepared the manuscript with the
contribution from all co-authors.

*Acknowledgements.* The authors would like to acknowledge the Department of Energy for funding this research under project name: Ocean_193360_1, project grant #: DE-SC0022103.



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
