# Peer review of "Investigating Lab-scaled Offshore Wind Aerodynamic Testing Failure and Developing Solutions for Early Anomaly Detections"

_Wind Energy Science, 2025_

## Author Comment (AC1)

**AC – 2nd Round of WES-2025-31 Manuscript Submitted on 01 Mar 2025**

We sincerely thank both reviewers for their thoughtful and constructive feedback. In response to their comments, we have made several substantial revisions aimed at improving the overall structure, clarity, and readability of the manuscript. The key modifications are summarized as follows:

- A new section titled "Problem Statement" has been introduced between the Methodology and Results sections. This section provides a clearer narrative by outlining the available experimental dataset, describing the general hyperparameter setup of the models, and presenting the rationale behind the selected anomaly detection criteria and their variations.

- The previously included subsection on "variation of anomaly detection criteria combination to the accuracy of the models" has been removed. The relevant results and insights are now integrated directly into the revised Results and Discussion sections for improved coherence.

- Tables 1 and 2 from the earlier manuscript have been consolidated and replaced with a single, more comprehensive table. This new table summarizes all models evaluated across different datasets and experimental conditions and is discussed in depth in the updated Discussion section.

- The sensitivity study subsection has been removed as a standalone section and its contents have been redistributed across relevant parts of the Results and Discussion sections.

- A new Results subsection has been added to present model performance under a series of synthetically introduced anomalies. This addition supports a broader evaluation of model generalization and robustness across different fault scenarios.
* * *
**RC1**
* * *
**1. Language and Grammar:**

**  - Par 25: Some sentences contain grammatical errors and awkward phrasing, making it challenging to understand the intended meaning. For instance, the use of the term "acute care" in a technical context is not entirely appropriate. I suggest replacing it with more suitable phrases such as "careful handling" or "close attention," depending on the tone you want.**

**A)** Thank you. The language is enhanced throughout the manuscript. Next is a screenshot of the delta file – explaining changes between first and second submission – with the specific change highlighted as requested).

To meet this demand, lab-scale turbine systems are designed to match the performance of full-scale offshore commercial wind plants , enabling accurate coupling between wind turbine aerodynamics and the hydrodynamic forces on the substructure Fowler et al. (2023); Kim (2014); Cao et al. (2023).  Due to the low Reynolds number at lab scale, thin airfoil sections are used for the model turbine blades, such as the SD7032, to achieve of the blades.  full scale rotor performance. However, this increases blade flexibility and reduces structural strength. Additionally, because of strict mass constraints-particularly for floating configurations-system redundancy that accommodate equipment malfunctions is typically not included in the design (Parker, 2022). As a result, lab-scale turbines are highly sensitive  systems ==that require careful handling== by operators to ensure safe and reliable operation throughout a test campaign.

**-There are a few instances where the text contains repetitive expressions, such as "as a consequence..." and "consequently," within the same paragraph. Reducing repetition will help enhance readability.**

**A)** We thank the reviewer for their comment. Here's an updated version:

In experimental testing campaigns,  particularly when testing novel control algorithms, the likelihood of fault events  ==consequences increases, and their impacts can be severe. These  ==consequences faults may arise from operator errors, incorrect control commands, or instrumentation malfunctions (Peng et al., 2023). Such  incidents can result in costly equipment damage, violations of laboratory safety standards, and substantial project delays. Therefore, efforts to develop efficient methods of detecting operational faults are critical to improving the testing process (Leahy et al., 2016; Lu et al., 2024).

**- Par 25. There are sentences that need to be fixed, for example, the sentence starting with "Furthermore" should read "Furthermore, due to tight mass considerations, particularly for floating models, system redundancy in the case of equipment malfunctions is generally not implemented, as noted by Parker (2022)."**

**A)** The overall language is improved as follows:

To meet this demand, lab-scale turbine systems are designed to match the performance of full-scale offshore commercial wind plants  , enabling accurate coupling between wind turbine aerodynamics and the hydrodynamic forces on the substructure Fowler et al. (2023); Kim (2014); Cao et al. (2023).  Due to the low Reynolds number at lab scale, thin airfoil sections are used for the model turbine blades, such as the SD7032, to achieve   full scale rotor performance. However, this increases blade flexibility and reduces structural strength. Additionally, because of strict mass constraints-particularly for floating configurations-system redundancy that accommodate equipment malfunctions is  typically not included in the design (Parker, 2022). As a result, lab-scale turbines are highly sensitive  systems that require careful handling by operators to ensure safe and reliable operation throughout a test campaign.

**- Par 150 is really difficult to understand (Especially the sentence starting with "The combinations..").  Both in terms of English and some parameters, such as DE|E and DE&E conditions, are not clearly explained.**

**A)** This is now further enhanced under the "Problem statement" section:

 Three combinations of anomaly detection criteria were investigated. The symbols $E$ and $\Delta E$ refer to threshold-exceeding conditions based on the model prediction error and its time derivative, respectively. The detection logic tested includes:

205

1. $\Delta E$ - the derivative of the error must exceed a threshold;

2. $\Delta E \vee E$ - either the error or its derivative must exceed its threshold.

3. $\Delta E \wedge E$ - both the error and its derivative must  simultaneously exceed their respective thresholds; and

210

We've added the: $\vee$ and $\wedge$ symbols to describe logical (or) and logical (and) mathematically.

**- Par 160: FOR is written in capital.  The sentence "As is...." Should be rewritten.**

**A)** This paragraph has been omitted.

**- Par 165: No verb in the first sentence.**
**A)** This paragraph has been omitted.
**- Par 190: Sentence "This work ...." should be rewritten.**

**A)** We thank the reviewer. The paragraph is rephrased as such:

This work serves as a proof of concept that  simple, interpretable, and computationally efficient techniques can be deployed to enhance safety and operational awareness during laboratory-scale wind turbine testing. The approach holds promise for extension to ocean-based and full-scale wind energy systems, where early anomaly detection is critical for preventing equipment failure and improving system reliability during experimental campaigns and operational phases.

**2. Clarity and Technical Accuracy:**

**- Par 35: In some sections, the explanation of velocity measurement as a vibration-based monitoring technique is not entirely clear. A more detailed rationale or supporting reference would improve comprehension.**

**A)** We have clarified the explanation of vibration-based monitoring techniques, particularly the use of velocity measurements, by referencing ISO 10816-21, which recommends evaluating vibration amplitudes using the root mean square (RMS) of velocity or acceleration signals. The paragraph has been revised accordingly for improved clarity, and additional supporting references have been incorporated to strengthen the discussion:

 Vibration-based condition monitoring techniques, often evaluated using the root mean square (RMS) of velocity or acceleration signals, are widely used for drivetrain fault detection, particularly to determine whether signal amplitudes exceed the thresholds defined by standards such as ISO 10816-21 (ISO, 1996). For example, Nejad et al. (2018) demonstrated that angular velocity measurements already available in existing control systems can be repurposed for fault detection, avoiding the need for costly additional instrumentation. Their approach was motivated by the challenge of identifying faults in  complex systems  with many interconnected components, where vibration signals may originate from various internal sources at different frequencies. In such cases, incorporating multiple  sensor channels is often necessary to obtain a more complete understanding of system behavior. However, this added complexity can increase computational cost and the risk of misinterpreting  irrelevant or noisy signal components.

**- Par 95: What does rotating a matrix mean? Do you mean transposing it? Needs to be clarified.**

**A)** This section is revised and further explained as follows:

Data are then standardized to ensure all channels (features)  have a mean value, $\mu$, of 0 and a standard deviation, $\sigma$, of 1:

$$x_i = \frac{x_i - \mu_i}{\sigma_i}, i = 1, ..., N,$$  (1)

where $N$ is the number of channels included in the model. Following standardization, the covariance matrix of the variables was computed and then diagonalized through eigendecomposition, yielding a set of orthogonal transformed variables—i.e., the principal components (PCs) ordered by the amount of total variance they explain. Based on this ranked structure, a subset of components can be selected to reduce the dimensionality of the problem while preserving as much of the original variance as possible. For instance, the first 5 PCs and channel loads/contributions to them is illustrated in Fig. 4b

**- Par 130: Information given here is mentioned earlier and is repeated here. This reduces the rigor, conciseness, and precision of the entire text.**

**A)** Rephrased

235  **4.3  Pre-strike anomaly detection**

During a high rotor angular velocity test, $\mathcal{D}_3$, an unexpected anomaly caused the rotor to accelerate rapidly. The resulting increase in thrust forces caused significant blade deflection, and within four seconds, one of the blades struck the tower, leading to severe damage, as shown in Figure 13.

**- Tables 1 and 2 have a central importance to the paper. But they are not adequately explained or referenced in the text. They should be explained and discussed thoroughly. Providing more context, especially when discussing essential results or comparisons, would enhance the reader's understanding.**

**A)** Agreed. First, a new table describing dataset usage for the models is discussed in Problem statement:

**Table 1.** Dataset usage by models $\mathcal{M}_1$ and $\mathcal{M}_3$ for different tasks. Time intervals are in seconds.

| Model | Task | Dataset(s) | Interval |
|---|---|---|---|
| $\mathcal{M}_1$ | Training | $\mathcal{D}_1$ | [100, 450] |
| | Validation | $\mathcal{D}_1$ | (450, 675] |
| | Error threshold | $\mathcal{D}_1$ | [100, 1000] |
| | Testing | $\mathcal{D}_2, \mathcal{D}_2^{(a1,a2,a3)}, \mathcal{D}_3$ | [100, 1000], [100, 350], (135, 190] |
| $\mathcal{M}_3$ | Training | $\mathcal{D}_3$ | [70, 119] |
| | Validation | $\mathcal{D}_3$ | (119, 135] |
| | Error threshold | $\mathcal{D}_3$ | [70, 135] |
| | Testing | $\mathcal{D}_3$ | (135, 190] |

This is important to help the reader understand the differences between the datasets used in the paper. Table 1 is further explained in the text in the problem statement section:

Three datasets, $\mathcal{D}_1$, $\mathcal{D}_2$, $\mathcal{D}_3$, were gathered during the test campaign. While wind speeds were kept constant (variation < 1%), the rotor angular velocity for $\mathcal{D}_2$ dataset was slightly lower by 12% and higher by 51% for $\mathcal{D}_3$, relative to $\mathcal{D}_1$. The angular velocity, and the resulting thrust force variations are illustrated in Fig. 9a and Fig. 9b, respectively. The blade pitch varied the same way for these cases based on the pre-generated setpoints. The actual anomaly and blade strike occurred near the end of $\mathcal{D}_3$, which was truncated to <200 s, while $\mathcal{D}_1$ and $\mathcal{D}_2$ each span 1000 s. In addition, three altered variants of $\mathcal{D}_2$ were generated to introduce a synthetic anomaly for further evaluation of the developed models: $\mathcal{D}_2^{(a1)}$, $\mathcal{D}_2^{(a2)}$, and $\mathcal{D}_2^{(a3)}$. The synthetic anomaly was imposed by modifying the tower base fore-aft bending moment, $M_y$, through a time-varying amplification factor. This factor was applied starting from an arbitrary onset time (225 s), increased linearly to a maximum value by 250 s, and then reduced back to unity by 275 s. The variants amplify the signal by 0.25%, 0.5%, and 1.00% per $\Delta t$ for $\mathcal{D}_2^{(a1)}$, $\mathcal{D}_2^{(a2)}$, and $\mathcal{D}_2^{(a3)}$, respectively. Table 1 summarizes the model setup and intervals of datasets utilized during training, validation, anomaly criteria threshold selection, and testing tasks.

After we performed all the analysis, we summarized it all in Table 2:

**Table 2.** Anomaly detection performance for models tested on datasets $\mathcal{D}_2$, $\mathcal{D}_2^{(a1)}$, $\mathcal{D}_2^{(a2)}$, $\mathcal{D}_2^{(a3)}$, and $\mathcal{D}_3$, under different detection criteria: $\Delta E$, $\Delta E \vee E$, and $\Delta E \wedge E$.

| Criterion | Dataset | 1PC | | | | | | | MPC | | | | | | |
|---|---|---|---|---|---|---|---|---|---|---|---|---|---|---|---|
| | | $T^+$ | $F^-$ | $F^+$ | $T^-$ | $P$ | $R$ | $FI$ | $T^+$ | $F^-$ | $F^+$ | $T^-$ | $P$ | $R$ | $FI$ |
| $\Delta E$ | $\mathcal{D}_2$ | 0 | 0 | 0 | 1800 | N/A | N/A | N/A | 0 | 0 | 0 | 1800 | N/A | N/A | N/A |
| | $\mathcal{D}_2^{(a1)}$ | 75 | 26 | 30 | 369 | 0.74 | 0.71 | 0.73 | 18 | 36 | 17 | 382 | 0.82 | 0.18 | 0.29 |
| | $\mathcal{D}_2^{(a2)}$ | 92 | 9 | 36 | 363 | 0.91 | 0.72 | 0.80 | 70 | 31 | 18 | 381 | 0.80 | 0.69 | 0.74 |
| | $\mathcal{D}_2^{(a3)}$ | 95 | 6 | 38 | 361 | 0.94 | 0.71 | 0.81 | 88 | 13 | 32 | 367 | 0.73 | 0.87 | 0.80 |
| | $\mathcal{D}_3$ | 49 | 2 | 4 | 125 | 0.96 | 0.93 | 0.94 | 30 | 21 | 0 | 129 | 1.00 | 0.59 | 0.74 |
| $\Delta E \vee E$ | $\mathcal{D}_2$ | 0 | 0 | 0 | 1800 | N/A | N/A | N/A | 0 | 0 | 0 | 1800 | N/A | N/A | N/A |
| | $\mathcal{D}_2^{(a1)}$ | 90 | 11 | 30 | 369 | 0.89 | 0.75 | 0.81 | 64 | 37 | 8 | 391 | 0.89 | 0.63 | 0.74 |
| | $\mathcal{D}_2^{(a2)}$ | 95 | 6 | 36 | 363 | 0.94 | 0.73 | 0.82 | 81 | 20 | 18 | 381 | 0.82 | 0.80 | 0.81 |
| | $\mathcal{D}_2^{(a3)}$ | 97 | 4 | 38 | 361 | 0.96 | 0.72 | 0.82 | 90 | 11 | 32 | 367 | 0.74 | 0.89 | 0.81 |
| | $\mathcal{D}_3$ | 51 | 0 | 129 | 0 | 1.00 | 0.28 | 0.44 | 51 | 0 | 121 | 8 | 0.30 | 1.00 | 0.46 |
| $\Delta E \wedge E$ | $\mathcal{D}_2$ | 0 | 0 | 0 | 1800 | N/A | N/A | N/A | 0 | 0 | 0 | 1800 | N/A | N/A | N/A |
| | $\mathcal{D}_2^{(a1)}$ | 65 | 36 | 21 | 378 | 0.64 | 0.76 | 0.70 | 18 | 83 | 1 | 398 | 0.95 | 0.18 | 0.30 |
| | $\mathcal{D}_2^{(a2)}$ | 87 | 17 | 28 | 371 | 0.83 | 0.75 | 0.79 | 65 | 36 | 17 | 382 | 0.79 | 0.64 | 0.71 |
| | $\mathcal{D}_2^{(a3)}$ | 89 | 12 | 32 | 367 | 0.88 | 0.74 | 0.80 | 81 | 20 | 25 | 374 | 0.76 | 0.80 | 0.78 |
| | $\mathcal{D}_3$ | 49 | 2 | 4 | 125 | 0.96 | 0.93 | 0.94 | 30 | 21 | 0 | 129 | 1.00 | 0.59 | 0.74 |

We explained the main findings thoroughly in the discussion session.

**5 Discussion**

 Table 2 summarizes the anomaly detection performance of model $\mathcal{M}_1$ with its 1PC and MPC variations, evaluated on the healthy dataset $\mathcal{D}_2$, synthetically altered anomaly datasets $\{\mathcal{D}_2^{(a1)}, \mathcal{D}_2^{(a2)}, \mathcal{D}_2^{(a3)}\}$, and the  blade-tower strike dataset $\mathcal{D}_3$. From these results, the following key observations can be made:

- The 1PC variation generally yields higher F1-scores compared to the MPC variation (43% enhancement under $\Delta E \wedge E$ criterion);

- The combined threshold criterion $\Delta E \wedge E$ provides the most consistent and reliable detection performance across datasets;

- While the 1PC model achieves higher recall ($R$), the MPC model tends to produce higher precision ($P$).

Importantly, both model variations produce no false positive detections under healthy conditions ($\mathcal{D}_2$), regardless of the threshold criterion employed. The 1PC model typically reacts more rapidly to actual anomalies, as it is not constrained by the

Additionally, we added a new figure that compares anomaly/non-anomaly events and the ability of the models to predict those events for all datasets tested and various anomaly conditions criteria:

[Figure]

**Figure 15.** Percentage of true positives, false negatives, false positive, and true negatives occurrence when testing $\mathcal{M}_1$ model for the variou testing datasets.

**- Uncertainty estimates for the P, R, and FI in Tables 1 and 2 would definitely help the technical rigor of the paper. For example, those parameters are given with 3 decimal resolution. Can this be justified?**

**A)** Uncertainty quantification was considered out of scope for this publication but an interesting point for future research discussion. There is no justification to the choice of decimals resolution selected (2 decimal resolution in the new manuscript).

**- Par 170: Increased frequency would help to make a faster detection, not reduced.**

**A)** Correct. This is now reflected in Figure 12.

[Figure]

**Figure 12.** Anomaly detection delay in seconds for $\mathcal{M}_1$ (both 1PC and MPC variations) when tested during various altered $\mathcal{D}_2$ datasets for two timestep realizations.

**3. Structural and Visual Presentation:**

**-The arrangement of the three graphs in Figure 9 can fit in a single line, which would help in better page management and improve readability.**

**A)** That figure has been removed and replaced with a simple version in Figure 12.

**4. Additional Considerations:**

**- It would be valuable to discuss the physical or mathematical reasons behind why the 1PC model performed better in anomaly detection.**

**A)** This is now further discussed in the discussion section. The newly introduced analysis (section 4.2 in the new manuscript) helped generalize the conclusion we reached. Also the discussion has shifted from saying 1PC is always performing better in anomaly detection into more detailed difference between the two variations. Namely, 1PC has higher recall, and therefore, earlier anomaly detection but that can lead to over-detection. MPC, on the other hand, is more conservative approach and has less recall but has higher precision. This, however, can lead to more delay in the anomaly recall and some detection latency. Therefore, each has advantages and disadvantages.

**1) Details and example of data-preprocessing with time-series signals, idle sensors and exact data cleaning method used for at least one representative case. What channels are used and why were they picked.**

**A)** Section 2.4 in the new manuscript provides a description of the numerous measurements and data channels used in the analysis presented in the paper as well as some data pre-processing that took place. For instance, we explained the standardization method for scaling, provide correlation matrix between channels of interest, and covariance loadings matrix after performing PCA: This is a screenshot of the delta file (difference between old and new manuscript):

**2.4 Anomaly detection over the span of multiple channels**

In complex systems such as offshore wind testing, there are numerous measurements and data channels, which can be used to understand the overall behavior of the system. However, for anomaly detection purposes, it can be overwhelming and computationally expensive to manually and in real-time search the data space for deviation in measured data. The operator might not have sufficient time to abort the test before the anomaly becomes too consequential. Additionally, an anomaly might not be detectable based on any single data channel to comprehend the full state of the system. Therefore, the predictive model must be based on multiple data channels related to the test being conducted while providing the operator with a single, concise, anomaly detection capability based on the most relevant information. To accomplish that, principal component analysis (PCA) was carried out. A PCA creates combinations of variables that explain the largest amount of variance in the data.

 Before performing the analysis, the raw data recorded by the data acquisition system  were pre-processed to remove idle measurements  and non-numeric  entries. Data channels collected comprises wind speed, angular velocity of the rotor, azimuth angle, all blades pitch angles, generator torque, rotor torque, forces and moments at the base of the tower. We assume the digital twin only has access to some of these channels (i.e., angular velocity, $\dot{\theta}$, rotor torque, $Q$, and tower base forces and moments: $F_x, F_y, F_z, M_x, M_y, M_z$) to simulate cases where some measurements can be restricted by turbine manufacturers and validate the model's operability under restrictive data access. Figure 4a illustrates the correlation matrix between the channels of interest.

Data are then standardized to ensure all channels (features)  have a mean value, $\mu$, of 0 and a standard deviation, $\sigma$, of 1:

$$x_i = \frac{x_i - \mu_i}{\sigma_i}, i = 1, ..., N, \tag{1}$$

where $N$ is the number of channels included in the model. Following standardization, the covariance matrix of the variables was computed and then diagonalized through eigendecomposition, yielding a set of orthogonal transformed variables—i.e., the principal components (PCs)  ordered by the amount of total variance  they explain. Based on this ranked structure, a subset of components can be selected to reduce the dimensionality of the problem while preserving as much of the original variance as possible. For instance, the first 5 PCs and channel loads/contributions to them is illustrated in Fig. 4b

The PCs  were then used to train the RNN model(s) that will later be used for prediction. As new data is acquired, it is  transformed/projected onto the same PCs that were used in training the models. For the purpose of anomaly detection, the mean absolute error (MAE) is computed between measurements and predictions from the RNN models, and the error derivative is calculated, to estimate rapid fluctuations in the quality of

(a) Correlation matrix between channels of interest.

(b) Covariance loadings matrix.

**Figure 4.** Data pre-processing: (a) correlation matrix between available channels used in the models, and (b) covariance loadings of the first 5 principal components.

We also provided full description of the pre-processing analysis used to determine threshold values for multiple principal component model (Figure 8 in the new manuscript). Additionally, in the new problem statement section, we explained in high detail the various dataset (cases) used in the analysis along with a description of the channels used in building the models and two plots that show time series variations in signals from datasets used in training/testing:

Three datasets, $\mathcal{D}_1$, $\mathcal{D}_2$, $\mathcal{D}_3$, were gathered during the test campaign. While wind speeds were kept constant (variation < 1%), the rotor angular velocity for $\mathcal{D}_2$ dataset was slightly lower by 12% and higher by 51% for $\mathcal{D}_3$, relative to $\mathcal{D}_1$. The angular velocity, and the resulting thrust force variations are illustrated in Fig. 9a and Fig. 9b, respectively. The blade pitch varied the same way for these cases based on the pre-generated setpoints. The actual anomaly and blade strike occurred near the end of $\mathcal{D}_3$, which was truncated to <200 s, while $\mathcal{D}_1$ and $\mathcal{D}_2$ each span 1000 s. In addition, three altered variants of $\mathcal{D}_2$ were generated to introduce a synthetic anomaly for further evaluation of the developed models: $\mathcal{D}_2^{(a1)}$, $\mathcal{D}_2^{(a2)}$, and $\mathcal{D}_2^{(a3)}$. The synthetic anomaly was imposed by modifying the tower base fore-aft bending moment, $M_y$, through a time-varying amplification factor. This factor was applied starting from an arbitrary onset time (225 s), increased linearly to a maximum value by 250 s, and then reduced back to unity by 275 s. The variants amplify the signal by 0.25%, 0.5%, and 1.00% per $\Delta t$ for $\mathcal{D}_2^{(a1)}$, $\mathcal{D}_2^{(a2)}$, and $\mathcal{D}_2^{(a3)}$, respectively. Table 1 summarizes the model setup and intervals of datasets utilized during training, validation, anomaly criteria threshold selection, and testing tasks.

~~During one of the high wind speed tests for the aerodynamic characterization of a 1: 50 scaled wind turbine, an erroneous activation of the emergency stop led to the shutdown of the generator. This caused the rotor to spin rapidly increasing thrust forces on the blades, causing them to bend more. Within less than four seconds, one of the blades struck the tower, resulting in severe damage as shown in figure 13~~Channels used in training the models include angular velocity, rotor torque, and tower base forces and moments. For most of the analyses presented in this paper, model $\mathcal{M}_1$ was employed. This model is trained on a previously available healthy dataset, $\mathcal{D}_1$, and serves as the primary reference. The rationale for this approach is based on the practical constraint that datasets containing anomalies rarely have a corresponding healthy segment recorded immediately beforehand. As such, training a model in real-time using only the healthy portion of a dataset that later exhibits an anomaly is typically infeasible. Nonetheless, for comparative purposes, we also evaluate model $\mathcal{M}_3$, which is trained on the healthy portion of dataset $\mathcal{D}_3$, under the hypothetical assumption that similar data had been recorded under identical conditions in advance.

**Table 1.** Dataset usage by models $\mathcal{M}_1$ and $\mathcal{M}_3$ for different tasks. Time intervals are in seconds.

| Model | Task | Dataset(s) | Interval |
|---|---|---|---|
| $\mathcal{M}_1$ | Training | $\mathcal{D}_1$ | [100, 450] |
| | Validation | $\mathcal{D}_1$ | (450, 675] |
| | Error threshold | $\mathcal{D}_1$ | [100, 1000] |
| | Testing | $\mathcal{D}_2$, $\mathcal{D}_2^{(a1,a2,a3)}$, $\mathcal{D}_3$ | [100, 1000], [100, 350], (135, 190] |
| $\mathcal{M}_3$ | Training | $\mathcal{D}_3$ | [70, 119] |
| | Validation | $\mathcal{D}_3$ | (119, 135] |
| | Error threshold | $\mathcal{D}_3$ | [70, 135] |
| | Testing | $\mathcal{D}_3$ | (135, 190] |

(a) Angular velocity of three experimental dataset.

[Figure]

(b) Thrust force measurements of three experimental dataset.

**Figure 9.** Three experimental datasets and their variations in (a) angular velocities and (b) thrust forces.

**2) Training - testing methodology: Type and size of RNN model used, time / resources needed for training.**

**A)** Section 3 in the new manuscript also provides full description now of the training/validation/testing methodology and the type, size, and hyperparameters of the models being used:

Models $\mathcal{M}_1$ and $\mathcal{M}_3$ were configured with identical training hyperparameters, except for the number of training epochs. Both models utilize a prediction horizon of a single timestep and a look-back to prediction ratio of $n/m = 10$. The network architecture consists of a single hidden layer with 100 neurons, trained using a batch duration of 60 seconds, a learning rate of 0.001, and no dropout regularization. Model $\mathcal{M}_1$ was trained for 60 epochs, whereas model $\mathcal{M}_3$ required an extended training schedule of 1000 epochs. This increase was motivated by the significantly shorter duration of training data available for $\mathcal{M}_3$, which spans only from 70 to 119 seconds due to the presence of an anomaly later in the dataset, as detailed in Table 1.

**Table 1.** Dataset usage by models $\mathcal{M}_1$ and $\mathcal{M}_3$ for different tasks. Time intervals are in seconds.

| Model | Task | Dataset(s) | Interval |
|---|---|---|---|
| $\mathcal{M}_1$ | Training | $\mathcal{D}_1$ | [100, 450] |
| | Validation | $\mathcal{D}_1$ | (450, 675] |
| | Error threshold | $\mathcal{D}_1$ | [100, 1000] |
| | Testing | $\mathcal{D}_2, \mathcal{D}_2^{(a1,a2,a3)}, \mathcal{D}_3$ | [100, 1000], [100, 350], (135, 190] |
| $\mathcal{M}_3$ | Training | $\mathcal{D}_3$ | [70, 119] |
| | Validation | $\mathcal{D}_3$ | (119, 135] |
| | Error threshold | $\mathcal{D}_3$ | [70, 135] |
| | Testing | $\mathcal{D}_3$ | (135, 190] |

**3) How many different faulty and non faulty cases are considered in simulation and how do 1PC and MPC compare in terms of false positive and negative detection. If this has not been studied yet, please add comments about this and/or add this as future work.**

**A)** We divided the results section to include:
1) testing under healthy dataset:

**4.1 Model performance during healthy conditions**

The performance of the $\mathcal{M}_1$ model, in terms of normalized error and error derivative to their respective threshold values, when tested against measured data during healthy operations, $\mathcal{D}_2$, are shown in Fig. 10. When using the lead principal component (i.e., 1PC variation of $\mathcal{M}_1$ model), the error values were consistent throughout the test. The MPC variation experienced a slight decline in error values as time progressed. As desired, both model variations exhibited no predicted anomalies based on any of the exceeding threshold criteria discussed.

[Figure]

(a) $\mathcal{M}_1$ 1PC model performance

(b) $\mathcal{M}_1$ MPC model performance

**Figure 10.** Error and error derivative curves between measured and $\mathcal{M}_1$ model during healthy $\mathcal{D}_2$ testing dataset when (a) a single or (b) multiple principal components are used.

2) testing under synthetically introduced anomalies:

[Figure]

(a) $\mathcal{M}_1$-1PC during $\mathcal{D}_2^{a1}$ dataset

(b) $\mathcal{M}_1$-1PC during $\mathcal{D}_2^{a2}$ dataset

(c) $\mathcal{M}_1$-1PC during $\mathcal{D}_2^{a3}$ dataset

(d) $\mathcal{M}_1$-MPC during $\mathcal{D}_2^{a1}$ dataset

(e) $\mathcal{M}_1$-MPC during $\mathcal{D}_2^{a2}$ dataset

(f) $\mathcal{M}_1$-MPC during $\mathcal{D}_2^{a3}$ dataset

**Figure 11.** Anomaly detection based on $\Delta E \wedge E$ criterion during synthetically altered dataset variations of $\mathcal{D}_2$ and $\mathcal{M}_1$ model detection response, with (1) 1PC - $\mathcal{D}_2^{(a1)}$, (b) 1PC - $\mathcal{D}_2^{(a2)}$, (c) 1PC - $\mathcal{D}_2^{(a3)}$, (d) MPC - $\mathcal{D}_2^{(a1)}$, (e) MPC - $\mathcal{D}_2^{(a2)}$, and (f) MPC - $\mathcal{D}_2^{(a3)}$ variations.

3) and the previous pre-strike anomaly case (actual anomaly):

[Figure]

(a) $\mathcal{M}_1$ model - 1PC  variation

(b) $\mathcal{M}_1$ model - MPC  variation

(c)  $\mathcal{M}_3$ model   variation

(d) $\mathcal{M}_3$ model - MPC variation

We then aggregated all these analysis in the discussion section into a single bar plot that shows all true positives, false negatives, false positives, and true negatives events for all these different cases:

[Figure]

**Figure 15.** Percentage of true positives, false negatives, false positive, and true negatives occurrence when testing $\mathcal{M}_1$ model for the various testing datasets.

This figure simply illustrates how the models were able to classify faulty and non-faulty conditions and how they compare to one another for various datasets tested:

across the same set of testing datasets. In the horizontal bar charts, darker shades correspond to the presence of anomalies in the data—hence their absence in the healthy dataset $\mathcal{D}_2$. The sign of each classification outcome indicates whether the model successfully detected an anomaly (positive) or failed to do so (negative). Color is used to convey prediction quality and context: green denotes correct classifications, while red indicates incorrect ones. This visual encoding effectively communicates both the correctness of model predictions and the operational context in which they occur, thereby emphasizing the model's ability to distinguish between healthy and anomalous system states.

The error derivative appears to be the dominant criterion for accurate anomaly detection. As shown in Fig. 15, the combined threshold criterion $\Delta E \wedge E$ results in fewer incorrect classifications (i.e., reduced red regions), whereas more flexible criteria—where either the error or its derivative ~~must be exceeded for the anomaly to be triggered. The precision and recall as well as the $FI$ score are presented. As the models were trained on healthy data, when being tested under other healthy data (with slight variations in blade pitch angle being tested), the HR, since it does not contain any anomalies, reports zero $T^+$ and $F^-$ values, but a non-zero $F^+$ value for $\Delta E$ and $\Delta E|E$ combinations because some normal samples were misclassified as anomalies. The $\Delta E\&E$ condition, however, does not trigger false anomalies and it also shows the highest $FI$ score. The MPC model shows similar patterns but with lower $FI$ scores.~~ alone exceeds the threshold—lead to increased misclassifications. Notably, the reconstruction error $E$ serves as a useful indicator for identifying deviations due to previously unseen operating conditions. In contrast, the error derivative $\Delta E$ is particularly effective in capturing abrupt transitions between the reconstructed and measured signals, making it well-suited for detecting sudden-onset anomalies such as the one present in this paper.

This analysis is also described numerically in Table 2:

**Table 2.** Anomaly detection performance for models tested on datasets $\mathcal{D}_2$, $\mathcal{D}_2^{(a1)}$, $\mathcal{D}_2^{(a2)}$, $\mathcal{D}_2^{(a3)}$, and $\mathcal{D}_3$, under different detection criteria: $\Delta E$, $\Delta E \vee E$, and $\Delta E \wedge E$.

| Criterion | Dataset | 1PC | | | | | | | MPC | | | | | | |
|---|---|---|---|---|---|---|---|---|---|---|---|---|---|---|---|
| | | $T^+$ | $F^-$ | $F^+$ | $T^-$ | $P$ | $R$ | $FI$ | $T^+$ | $F^-$ | $F^+$ | $T^-$ | $P$ | $R$ | $FI$ |
| $\Delta E$ | $\mathcal{D}_2$ | 0 | 0 | 0 | 1800 | N/A | N/A | N/A | 0 | 0 | 0 | 1800 | N/A | N/A | N/A |
| | $\mathcal{D}_2^{(a1)}$ | 75 | 26 | 30 | 369 | 0.74 | 0.71 | 0.73 | 18 | 36 | 17 | 382 | 0.82 | 0.18 | 0.29 |
| | $\mathcal{D}_2^{(a2)}$ | 92 | 9 | 36 | 363 | 0.91 | 0.72 | 0.80 | 70 | 31 | 18 | 381 | 0.80 | 0.69 | 0.74 |
| | $\mathcal{D}_2^{(a3)}$ | 95 | 6 | 38 | 361 | 0.94 | 0.71 | 0.81 | 88 | 13 | 32 | 367 | 0.73 | 0.87 | 0.80 |
| | $\mathcal{D}_3$ | 49 | 2 | 4 | 125 | 0.96 | 0.93 | 0.94 | 30 | 21 | 0 | 129 | 1.00 | 0.59 | 0.74 |
| $\Delta E \vee E$ | $\mathcal{D}_2$ | 0 | 0 | 0 | 1800 | N/A | N/A | N/A | 0 | 0 | 0 | 1800 | N/A | N/A | N/A |
| | $\mathcal{D}_2^{(a1)}$ | 90 | 11 | 30 | 369 | 0.89 | 0.75 | 0.81 | 64 | 37 | 8 | 391 | 0.89 | 0.63 | 0.74 |
| | $\mathcal{D}_2^{(a2)}$ | 95 | 6 | 36 | 363 | 0.94 | 0.73 | 0.82 | 81 | 20 | 18 | 381 | 0.82 | 0.80 | 0.81 |
| | $\mathcal{D}_2^{(a3)}$ | 97 | 4 | 38 | 361 | 0.96 | 0.72 | 0.82 | 90 | 11 | 32 | 367 | 0.74 | 0.89 | 0.81 |
| | $\mathcal{D}_3$ | 51 | 0 | 129 | 0 | 1.00 | 0.28 | 0.44 | 51 | 0 | 121 | 8 | 0.30 | 1.00 | 0.46 |
| $\Delta E \wedge E$ | $\mathcal{D}_2$ | 0 | 0 | 0 | 1800 | N/A | N/A | N/A | 0 | 0 | 0 | 1800 | N/A | N/A | N/A |
| | $\mathcal{D}_2^{(a1)}$ | 65 | 36 | 21 | 378 | 0.64 | 0.76 | 0.70 | 18 | 83 | 1 | 398 | 0.95 | 0.18 | 0.30 |
| | $\mathcal{D}_2^{(a2)}$ | 87 | 17 | 28 | 371 | 0.83 | 0.75 | 0.79 | 65 | 36 | 17 | 382 | 0.79 | 0.64 | 0.71 |
| | $\mathcal{D}_2^{(a3)}$ | 89 | 12 | 32 | 367 | 0.88 | 0.74 | 0.80 | 81 | 20 | 25 | 374 | 0.76 | 0.80 | 0.78 |
| | $\mathcal{D}_3$ | 49 | 2 | 4 | 125 | 0.96 | 0.93 | 0.94 | 30 | 21 | 0 | 129 | 1.00 | 0.59 | 0.74 |

It is very interesting to see how the two models vary in terms of recall and precision. Namely, 1PC has higher recall, and therefore, earlier anomaly detection but that can lead to over-detection. MPC, on the other hand, is more conservative approach and has less recall but has higher precision. This, however, can lead to more delay in the anomaly recall and some detection latency. Therefore, each has advantages and disadvantages. We are, therefore, shifting our language from specifically saying one model is better than the other and leaving it open to discussion. The newly introduced analysis (section 4.2 in the new manuscript) helped generalize the conclusion we reached.

---

## Author Response (AR2)

**Response to Report #2**

This work proposes a PCA-LSTM anomaly detection approach for a lab-scale wind turbine failure detection. While the methodology is sound and demonstrates clear value, several technical details require clarification for improved reproducibility.

1) The discussion of vibration-based condition monitoring in introduction seems misplaced since the approach used angular velocity, torque, and force measurements rather than vibration signals. Please clarify the relevance or consider focusing the literature review on multi-sensor anomaly detection methods more aligned with your methodology.

We appreciate the reviewer's comment on this. Our intention was not to emphasize vibration-based condition monitoring techniques but rather to highlight that effective fault detection can be achieved using measurements already available in the system, such as angular velocity or torque measurements, without the need for additional sensors. We have revised the paragraph in the introduction to clarify this point and shifted the focus from vibration-based sensors and clarified our emphasis on multi-sensor anomaly detection methods that leverage existing control system data. We have added another citation regarding multi-sensor fusion for anomaly detection that aligns more directly with the methodology used in this study.

Vibration-based condition monitoring In many condition monitoring applications, anomaly detection is performed using dedicated sensors. For instance, vibration-based techniques, often evaluated using the root mean square (RMS) of velocity or acceleration signals, are widely used for drivetrain fault detection, particularly to determine whether signal amplitudes exceed the thresholds defined by and are assessed against standards such as ISO 10816-21 (ISO, 1996). For example, However, deploying additional instrumentation is not always feasible or cost-effective. Nejad et al. (2018) demonstrated that angular velocity measurements already available in within existing control systems can be repurposed for fault detection, avoiding thereby eliminating the need for costly additional instrumentation. Their approach was motivated by the challenge of identifying faults in complex systems with many interconnected components, where vibration signals may originate from various internal sources at different frequencies. In such cases, incorporating multiple sensor channels is often necessary to obtain a more complete understanding of system behavior. However, this added complexity can increase computational cost and the risk of misinterpreting irrelevant or noisy signal components, supplementary sensors. Similarly, Dameshghi and Refan (2019) proposed a diagnostic approach for gearbox faults based on SCADA information multi-sensor fusion, avoiding the need for additional data collection systems. These approaches illustrate the potential of multi-sensor anomaly detection methods that leverage existing system measurements.

2) Please show absolute values in both correlation and covariance matrices in figure 4 for clearer interpretation.

Figure 4 has been updated to show the absolute values

3&4) Please maintain consistent terminology throughout the paper. Authors refer to RNN in lines 113-115 but use LSTM elsewhere. Please consider using LSTM consistently to avoid confusion. Why choose MAE over MSE for reconstruction error? MAE can be less sensitive to outliers, but MSE might better capture the magnitude of deviations. Please justify this choice.

Thank you for pointing this important distinction. We chose the mean absolute error (MAE) over the mean squared error (MSE) as our reconstruction error to reduce the model's sensitivity to transient spikes or noise that may not necessarily correspond to actual anomalies. This allows the model to only detect sustained system variations from nominal behavior. With that said, we will consider incorporating a comparative analysis in possible future extensions of this work to further asses the impact of the choice of reconstruction error metric. In the meantime, we have added a justification to the use of MAE in the manuscript as follows:

The PCs were then used to train the RNN model(s) LSTM models that will later be used for prediction. As new data is acquired, it is transformed/projected onto the same PCs that were used in training the models. For the purpose of anomaly detection, the mean absolute error (MAE) is computed between measurements and predictions from the RNN-LSTM models, and the error derivative is calculated, to estimate rapid fluctuations in the quality of the predictions. The choice of MAE as the reconstruction error metric was made to reduce model sensitivity to transient spikes or noise, which may not correspond to true anomalies. An anomaly alert is reported to the operator when certain anomalous conditions are met. In this research, we investigate conditions when both the error and its derivative were crossing certain thresholds. This procedure is illustrated in Figure 5 and is explained in section 2.6.

5) Please specify the input sequence length for the LSTM model.

The input sequence length is specified in the manuscript as a look-back to prediction ratio of n/m=10, corresponding to a look-back window (input sequence length) of 10 timesteps for a 1-step prediction horizon. To improve clarity, we have now explicitly stated the input sequence length in the text.

Models  $\mathcal{M}_1$  and  $\mathcal{M}_3$  were configured with identical training hyperparameters, except for the number of training epochs. Both models utilize a prediction horizon of a single timestep and a look-back to prediction ratio of n/m = 10, corresponding to an input sequence length of 10 timesteps. The network architecture consists of a single hidden layer with 100 neurons,

6) Given the standardization approach in equation 1 where data is normalized to mean zero and standard deviation of one, and considering that synthetic anomalies are created by amplifying monitored signals, a fundamental question arises: if both training healthy data and anomalous test data are normalized to the same scale, how can the resulting deviations be detected by the model? Please clarify in this regard.

We appreciate the reviewer's thoughtful question. To clarify, the standardization procedure described in Equation 1 is applied using the mean and standard deviation computed from the healthy training dataset **only**. The resulting scaler is then stored and applied to transform all subsequent input data (could be healthy or anomalous), prior to PCA projection (the PCA transform, not to be confused with scaler transform) and prediction. This ensures that deviation from the distribution of the healthy training data are preserved and detectable. We have updated the manuscript to explicitly state that the standardization model (scaler) is fitted once on the training data and reused for transforming new data.

Data are then standardized to ensure all channels (features) in the training dataset have a mean value,  $\mu$ , of 0 and a standard deviation,  $\sigma$ , of 1:

$$x_i = \frac{x_i - \mu_i}{\sigma_i}, i = 1, ..., N,$$
 (1)

where N is the number of channels included in the model. All subsequent testing dataset (whether healthy or anomalous) are standardized using these scaler parameters. This ensures that the resulting transformed values might reflect deviations from the training dataset, allowing the model to identify anomalous behavior.

7) According to figure 11, it seems 1PC outperforms MPC despite MPC containing all information from 1PC plus additional components. Please investigate whether certain channels hinder rather than help anomaly detection, as this would strengthen the proposed methodology's motivation.

We believe the reviewer is raising an interesting point of view. In the manuscript, we clarified that the anomaly was introduced in the tower base signal, which has a relatively strong loading in PC1. This likely contributed to 1PC performing better than MPC despite having MPC containing all information from 1PC plus additional components. This is because this added information can dilute the influence of specific anomalous channels, especially when the anomaly is strongly represented in the leading component but has minimal contributions in subsequent components. Conversely, if an anomaly were introduced in a channel with weak or near-zero loading in PC1, its detection would likely require the inclusion of additional components. Thus, while MPC offers broader coverage across the feature space, it may also distribute the reconstruction error in a way that reduces sensitivity to certain localized anomalies. Therefore, the use of additional PCs

may become necessary to capture anomalies in channels that do not have high loading in PC. Deeper investigation into how channel loadings influence anomaly detectability is an interesting direction for future research.

That said, we also note that in a physically coupled system such as the one studied here, localized anomalies may inherently manifest across multiple correlated channels due to system dynamics. As a result, even anomalies originating in channels with low PC1 loadings could still influence leading components via cross-correlations.

The 1PC variation demonstrates overall enhanced coverage (highlighted in blue) and reduced detection delay relative to the onset of the ground-truth anomaly (highlighted in red). The reduction in detection delay is also represented This is particularly evident in Fig. 12 when comparing which compares detection delays for 1PC to MPC and MPC under various anomaly

12

Figure 11. Anomaly detection based on  $\Delta E \wedge E$  criterion during synthetically altered dataset variations of  $\mathcal{D}_2$  and  $\mathcal{M}_1$  model detection response, with (1) 1PC -  $\mathcal{D}_2^{(a1)}$ , (b) 1PC -  $\mathcal{D}_2^{(a2)}$ , (c) 1PC -  $\mathcal{D}_2^{(a3)}$ , (d) MPC -  $\mathcal{D}_2^{(a1)}$ , (e) MPC -  $\mathcal{D}_2^{(a2)}$ , and (f) MPC -  $\mathcal{D}_2^{(a3)}$  variations.

scenarios. Although the MPC model includes more principal components, this added information can dilute the influence of specific anomalous channels, especially when the anomaly is strongly represented in the leading component but has minimal contributions in subsequent components. Conversely, if an anomaly were introduced in a channel with weak or near-zero loading in PC1, its detection would likely require the inclusion of additional components. Thus, while MPC offers broader coverage across the feature space, it may also distribute the reconstruction error in a way that reduces sensitivity to certain localized anomalies. Additionally, detection performance generally improved with increasing severity of the synthetically introduced anomaly. This is indicated in Fig. 12 which shows detection delay in seconds between synthetically introduced anomaly and the predicted anomaly by the models. The figure also shows a sensitivity analysis of the models to the timestep at which the data is sampled. Small timesteps (high sampling frequency) can provide reduce anomaly detection delay but at the expense of computational cost.